# Immunogenicity of Alternative Dosing Schedules for HPV Vaccines among Adolescent Girls and Young Women: A Systematic Review and Meta-Analysis

**DOI:** 10.3390/vaccines8040618

**Published:** 2020-10-20

**Authors:** Andrew M. Secor, Matthew Driver, Brenda Kharono, Dianna Hergott, Gui Liu, Ruanne V. Barnabas, Peter Dull, Stephen E. Hawes, Paul K. Drain

**Affiliations:** 1START Center, University of Washington, Seattle, WA 98195, USA; mpdriver@uw.edu (M.D.); bkharono@uw.edu (B.K.); dhergott@uw.edu (D.H.); hawes@uw.edu (S.E.H.); pkdrain@uw.edu (P.K.D.); 2Department of Global Health, University of Washington, Seattle, WA 98195, USA; rbarnaba@uw.edu; 3Department of Epidemiology, University of Washington, Seattle, WA 98195, USA; guiliu@uw.edu; 4Department of Medicine, University of Washington, Seattle, WA 98195, USA; 5Bill and Melinda Gates Foundation, Seattle, WA 98109, USA; peter.dull@gatesfoundation.org

**Keywords:** HPV, vaccines, immunogenicity, alternative dosing schedules, meta-analysis

## Abstract

Alternative dosing schedules for licensed human papilloma virus (HPV) vaccines, particularly single dose and extended intervals between doses (>12 months), are being considered to address vaccine shortages and improve operational flexibility. We searched PUBMED/MEDLINE for publications reporting immunogenicity data following administration of one of the licensed HPV vaccines (2vHPV, 4vHPV, and 9vHPV) to females aged 9–26 years. We conducted non-inferiority analyses comparing alternative to standard schedules using mixed effects meta-regression controlling for baseline HPV status and disaggregated by vaccine, subtype, time point, and age group (9–14 and 15–26 years). Non-inferiority was defined as the lower bound of the 95% confidence interval (CI) for the geometric mean titer (GMT) ratio being greater than 0.5. Our search returned 2464 studies, of which 23 were included in data analyses. When evaluated against standard schedules, although robust immunogenicity was demonstrated across all multi-dose groups, non-inferiority of extended interval dosing was mixed across vaccines, subtypes, and time points. Single dose did not meet the criteria for non-inferiority in any comparisons. Sparse data limited the number of possible comparisons, and further research is warranted.

## 1. Introduction

Cervical cancer is one of the most common cancers among women [1]. There are an estimated 570,000 incident cases per year, and in 2018 an estimated 311,000 women died from the disease [2]. Human papilloma virus (HPV) is the leading cause of cervical cancer, with the majority of cases attributable to subtypes 16 and 18 [3]. HPV is one of the most common sexually transmitted infections, with a lifetime risk of infection of 75–80% among sexually active individuals [4] and an estimated global prevalence of 11–12% [5]. Higher rates of both HPV infection and cervical cancer have been seen in low- and middle-income (LMIC) countries [2,5], and it is estimated that LMICs account for 90% of cervical cancer deaths [6].

Three HPV vaccines licensed by the Food and Drug Administration (FDA) and European Medicines Agency (EMA) were approved for a three-dose schedule (0, 1 or 2, 6 months). After several clinical trials demonstrated non-inferiority in non-immunocompromised individuals between ages 9 and 14, the WHO recommended a two-dose schedule (0, 6 months for the quadrivalent and nonavalent vaccines, and 0, 5–13 months for the bivalent vaccine) [7]. In response to global vaccine shortages, and to promote greater vaccination rates by decreasing barriers to schedule completion and costs of implementation of vaccination programs, other alternative dosing schedules are now being explored [8,9,10]. Specifically, extended interval dosing, defined here as at least 12 months between first and second dose, and single dosing are being investigated as ways to increase vaccination rates, particularly in LMICs. Extended interval dosing could increase schedule completion by allowing more flexibility for follow-up visits, and a single dose schedule would allow for higher coverage with reduced costs, which is particularly important given the substantially higher cost of HPV vaccines as compared to other vaccines.

Current single dose data are limited to post-hoc analyses, and there are limited extended interval dosing data. On-going randomized controlled trials (RCTs) will deliver immunogenicity and efficacy data on single dose in the coming years, with the novel Cecolin vaccine already showing promising results [9]. The aim of this study was to systematically review and compare the existing data on the immunogenicity of extended interval and single dose schedules versus standard dose schedules in a non-inferiority meta-analysis, in order to detail and assess the current evidence base of these alternative dosing schedules.

## 2. Materials and Methods

### 2.1. Literature Search and Study Selection

In accordance with PRISMA guidelines [11], we conducted a systematic review to identify post-vaccination immunogenicity data for HPV vaccines among human participants. We searched the PUBMED/MEDLINE database for English-language publications indexed through 29 April 2019. Abstracts and full-text publications were screened independently using Excel and Covidence software (Covidence, Melbourne, Australia). All publications were screened by at least two reviewers, and conflicts were resolved via a third-party reviewer.

Inclusion criteria were: human studies reporting geometric mean titers (GMT) for HPV subtypes 16 and 18 of adolescent girls and young women aged 9–26 years following vaccination with one of the three licensed HPV vaccines using either standard, extended, or single dosing schedules via intramuscular administration. Studies meeting those criteria were excluded if they reported results in units without published conversion factors for international units. For studies in which GMT estimates or 95% confidence intervals were not explicitly reported in the text of the publication, relevant data were requested from the corresponding authors.

### 2.2. Data Extraction

Two reviewers extracted and verified the data, and a third reviewer resolved any conflicts. We abstracted study metadata, including: title, author, publication year, PMID number, NCT number, assay type, vaccine type and manufacturer, HPV type tested or reported (i.e., HPV subtype 16 or 18), sex, age, sample size, baseline HPV status, country, HIV status, presence of other comorbidities, number of doses received before antibody testing, dosing schedule, time between receipt of first vaccine and antibody testing, GMT point estimates, 95% confidence intervals, and units.

We extracted GMTs reported at one month post-last dose, 36 months post-first dose, and 72 months post-first dose in order to evaluate short-term, mid-term, and long-term immunogenicity, respectively. Data reported within 12 months of the mid-term time point or within 24 months of the long-term time point were collected if 36-month and 72-month data were not available, respectively; for example, if a study reported GMTs at month 30 but not month 36, the month 30 GMT would be extracted for the mid-term time point.

All comparisons and pooled estimates were calculated using international units (IU/mL). Conversion rates for mMU/mL and EU/mL to IU/mL were obtained from published literature [12,13].

### 2.3. Definitions

#### 2.3.1. Vaccines

There are currently three HPV vaccines licensed by the Food and Drug Administration (FDA) and European Medicines Agency (EMA): Bivalent (HPV 16/18, Cervarix, GSK), quadrivalent (HPV 6/11/16/18, Gardasil-4, Merck), and nonavalent (HPV 6/11/16/18/31/33/45/52/58, Gardasil-9, Merck), all of which have been shown to be highly efficacious [7,14,15,16,17,18,19].

#### 2.3.2. Age

Adolescent girls and young women aged 9–26 years were included in the study. Ages were aggregated in the following ranges to match current HPV vaccine policy and in response to differential immunogenicity by age group: 9–14 and 15–26. Studies with reported data aggregated across our age groups (e.g., ages 10–18) were excluded, with one exception: Gilca 2019 [20], a key source of single and extended dose data, which reported aggregated results for ages 13–18. As prior research shows that younger groups tend to have higher immunogenicity [21], these data were coded as 9–14 for a more conservative estimate of immune response in alternative schedules.

#### 2.3.3. Dose

Extended dose was defined as at least 12 months between doses. Standard dose was defined by current HPV vaccination policy: 2 doses at months 0 and 6, or 3 doses at months 0, 1 or 2, and 6 [7].

#### 2.3.4. Analysis

GMT ratios and confidence intervals (CIs) comparing alternative to standard schedules were calculated using mixed effects meta-regression controlling for baseline HPV status and disaggregated by vaccine, HPV subtype, time point, and age group (9–14 and 15–26 years). Due to a lack of an established correlate of protection for HPV vaccine immunogenicity, we conducted a non-inferiority analysis to compare alternative to standard dosing schedules. Non-inferiority was assessed by estimating the ratio of pooled GMTs comparing extended interval or single dose to standard schedules at the same time point, with non-inferiority defined as the lower bound of the 95% CI for the GMT ratio being greater than 0.5 [22]. All analyses were conducted in R (version 3.6.1, R Foundation for Statistical Computing, Vienna, Austria).

## 3. Results

### 3.1. Study Selection

We identified 2464 studies using the search criteria (Figure 1). After removing duplicates and screening titles and abstracts, 367 studies were included for full-text review. Of these, 344 were excluded from the present analysis. The most common reasons for exclusion were lack of relevant immunogenicity data (107), duplicate or previously reported data (54), using an assay without a published conversion factor for international units (17), or trials using a non-licensed vaccine (16). Of note, three single dose and/or extended interval studies were excluded from the meta-analysis, two due to using an assay without a published conversion factor for international units [23,24], and one due to lack of available GMT data [25]. Standard dose studies were excluded if there were no available alternative schedule comparison data.

In total, 23 studies were included in the present analysis; four contributed to extended interval data only [26,27,28,29], two to single dose data only [10,30], one study contributed to both [20], and 16 to comparator data only [17,19,21,22,31,32,33,34,35,36,37,38,39,40,41,42].

### 3.2. Study Characteristics

Included alternative dose studies are detailed in Table 1. All three licensed vaccines were included in the analysis. Extended interval had data for all three time points, while single dose only had mid- (36 months) and long-term (72 months) data. Included studies were conducted in North America (13), South America (2), Europe (9), Africa (3), and Asia (10). Seventeen of the included studies were randomized control trials, five were non-randomized trials or cohort studies, and one was a cross-sectional study. Sample sizes ranged from 18 to 2635.

### 3.3. Extended Interval

Five studies contributed to the extended interval data and 17 to the comparison data, including four contributing to both. Extended interval data were only available for the 9–14 age range for all three vaccines. The bivalent vaccine had data for one month and 36 months post-first dose, the quadrivalent vaccine had data for all three time points, and the nonavalent vaccine for only one month post-last dose. Figure 2 shows comparisons in pooled GMT, and Figure 3 shows GMT ratios and the non-inferiority analysis.

#### 3.3.1. Bivalent

Two studies reported bivalent vaccine extended dose data: Puthanakit 2016 (one month) [29] and Huang 2017 (36 months) [26]. Extended interval demonstrated non-inferiority for the bivalent vaccine for HPV 18 at one month post-last dose and 36 months post-first dose. Non-inferiority was not met for HPV 16 at either of those time points in this vaccine.

HPV 16 GMT estimates for standard vs. extended dose were 1851 IU/mL (95% CI 1269–2699) vs. 1569 IU/mL (95% CI 712–3455) at one month post-last dose, and 244 IU/mL (95% CI 155–383) vs. 214 IU/mL (95% CI 89–512) at 36 months. HPV 18 GMT estimates for standard vs. extended dose were 1247 IU/mL (95% CI 973–1599) vs. 1251 IU/mL (95% CI 773–2026) at one month post-last dose, and 127 IU/mL (95% CI 88–185) vs. 151 IU/mL (95% CI 74–308) at 36 months.

#### 3.3.2. Quadrivalent

Three studies reported quadrivalent vaccine extended dose data: Gilca 2019 (one month) [20], LaMontagne 2013 (one month and 72 months) [27], and Neuzil 2011 (36 months) [28]. Extended interval demonstrated non-inferiority in the quadrivalent vaccine for HPV 18 at all time points and for HPV 16 at 36 months and 72 months. Non-inferiority was not met for HPV 16 at one month post-last dose.

HPV 16 GMT estimates for standard vs. extended dose were 1678 IU/mL (95% CI 882–3193) vs. 1474 IU/mL (95% CI 781–2779) at one month post-last dose, 329 IU/mL (95% CI 91–1189) vs. 297 IU/mL (95% CI 82–1080) at 36 months, and 167 IU/mL (95% CI 97–286) vs. 248 IU/mL (95% CI 146–422) at 72 months. HPV 18 GMT estimates for standard vs. extended dose were 297 IU/mL (95% CI 189–467) vs. 297 IU/mL (95% CI 178–496) at one month post-last dose, 39 IU/mL (95% CI 18–84) vs. 28 IU/mL (95% CI 14–57) at 36 months, and 15 IU/mL (95% CI 8–28) vs. 28 IU/mL (95% CI 16–52) at 72 months.

#### 3.3.3. Nonavalent

One study reported nonavalent vaccine extended dose data: Gilca 2019 (one month) [20]. Extended interval was non-inferior to the standard dose schedule for both HPV 16 and HPV 18 at one month post-last dose.

At one month post-last dose, HPV 16 GMT estimates for standard vs. extended dose were 1525 IU/mL (95% CI 1081–2152) vs. 1717 IU/mL (95% CI 869–3392), and HPV 18 GMT estimates were 311 IU/mL (95% CI 215–451) vs. 396 IU/mL (95% CI 201–781).

### 3.4. Single Dose

Three studies contributed to the single dose data and 14 to the comparison data, including two contributing to both. Single dose data were only available for the bivalent and quadrivalent vaccines, both of which had data at 36 months and 72 months post-first dose. Quadrivalent vaccine data were limited to the 9–14 age range, while bivalent vaccine data were available for both 9–14 and 15–26 ranges (although not at all time points). Figure 4 shows comparisons in pooled GMT, and Figure 5 shows GMT ratios and the non-inferiority analysis. Non-inferiority of single dose compared to standard dose was not demonstrated for either vaccine for both subtypes in relation to all time points and age groups compared.

#### 3.4.1. Bivalent

Two studies reported bivalent vaccine single dose data: LaMontagne 2014 (36 months, 9–14 years) [30] and Safaeian 2018 (36- and 72-month data, 15–26 years) [10]. Non-inferiority was not met for either HPV sub-type for any of the age groups or time points.

For the 9–14 year old group, at 36 months post-first dose, HPV 16 GMT estimates for standard vs. single dose were 244 IU/mL (95% CI 155–383) vs. 31 IU/mL (95% CI 9–111), and HPV 18 GMT estimates were 127 IU/mL (95% CI 88–185) vs. 16 IU/mL (95% CI 5–49).

Among 15–26 year olds, HPV 16 GMT estimates for standard vs. single dose were 102 IU/mL (95% CI 70–149) vs. 28 IU/mL (95% CI 15–54) at 36 months, and 74 IU/mL (95% CI 54–102) vs. 27 IU/mL (95% CI 14–50) at 72 months. HPV 18 GMT estimates for standard vs. single dose were 66 IU/mL (95% CI 44–98) vs. 21 IU/mL (95% CI 12–39) at 36 months, and 50 IU/mL (95% CI 37–69) vs. 24 IU/mL (95% CI 13–42) at 72 months.

#### 3.4.2. Quadrivalent

One study reported quadrivalent vaccine single dose data: Gilca 2019 (36- and 72-months, 9–14 years) [20]. Non-inferiority was not met for either HPV sub-type for any of the age groups or time points.

HPV 16 GMT estimates for standard vs. extended dose were 329 IU/mL (95% CI 91–1189) vs. 18 IU/mL (95% CI 3–102) at 36 months post-first dose, and 167 IU/mL (95% CI 97–286) vs. 22 IU/mL (95% CI 6–87) at 72 months. HPV 18 GMT estimates for standard vs. extended dose were 39 IU/mL (95% CI 18–84) vs. 6 IU/mL (95% CI 1–23) at 36 months post-first dose, and 15 IU/mL (95% CI 8–28) vs. 8 IU/mL (95% CI 2–32) at 72 months.

#### 3.4.3. Nonavalent

No single dose data were available for the nonavalent vaccine.

## 4. Discussion

We analyzed data from 23 published studies comparing alternative dosing schedules (extended interval or single dose) to standard dosing schedules. Non-inferiority analyses showed mixed results based on dosing regimen, vaccine type, and time point. Our analysis utilized GMT levels as a correlate of protection. However, there is no established threshold of HPV titer that indicates protection; therefore, not meeting the non-inferiority criteria does not necessarily mean that the alternative schedules will not confer adequate protection against HPV 16 and 18 infection.

For extended interval analyses, non-inferiority was demonstrated for all three vaccines at multiple time points for both HPV 16 and HPV 18 titers, with the exception of HPV 16 titers of the bivalent vaccine at one month post-last dose and 36 months post-first dose, and HPV 16 titers for the quadrivalent vaccine one month post-last dose. Interestingly, Puthanakit et al. [29], one of the studies contributing to our analysis, found a non-inferior relationship between extended dose and standard schedules for the bivalent vaccine at one month post-last dose. This difference in findings from our own analysis is likely due to our pooling of results (six studies, including Puthanakit et al., contributed standard dose data for the bivalent, one-month extended interval analysis), which altered the comparison group, such as differing age structures. While Puthanakit et al. included ages 9–14, nearly half of the pooled comparison group came from a study that was restricted to ages 9–10 [22]. As younger participants often show higher immune responses, this may have resulted in a higher titer level among the pooled standard dose group as compared to the Puthanakit et al. analysis. The difference could also be due to random variation.

For single dose regimens, non-inferiority was not demonstrated for bivalent or quadrivalent vaccines at any time point in our studies. However, the data on single dose vaccinations were limited and none came from studies designed to assess single dose immunogenicity; all three included studies collected single dose data from participants who were supposed to receive 2 or 3 doses but failed to do so. As these individuals were not randomized to receive the single dose schedule, these results may not be generalizable to larger populations. In addition, as these data are from individuals who were supposed to receive additional doses, it was not possible to assess whether a single vaccination with a higher dose could confer a greater immunological response. There are a number of ongoing RCTs of single dose schedules that are attempting to answer these questions (see clinical trial registrations NCT03180034, NCT03675256, NCT03832049, and NCT02834637).

Our analysis suggests that extended interval dosing schedules may be an effective alternative for HPV vaccination. Under the current schedules, both the market availability and logistical hurdles make it difficult for countries to achieve high coverage. A 2016 systematic review of global HPV vaccine coverage found that coverage varies greatly by region and income level [43]. The review found that globally, 6.1% of women aged 10–20 had received a full course of HPV vaccination. However, most vaccine recipients were from high-income countries, with 33.6% coverage in developed countries compared to 2.6% in developing countries. Africa and Asia both showed 1.2% vaccine coverage of this target demographic. If extended interval dosing provides similar amounts of protection as the currently recommended schedules, as our analysis suggests, this could improve vaccine coverage and reduce the barriers to access, especially in LMICs.

Extended interval and single dosing alternative schedules may provide several benefits. Single dose vaccination would decrease the cost of nationwide HPV vaccination programs, reducing logistics, supply chain, and staffing expenses. Eliminating or allowing for greater flexibility in the timing of the second dose would also decrease the programmatic challenges of vaccine delivery and reduce the burden on clients to attend clinic visits to complete the vaccine schedule. Extended interval dosing would also simplify school-based immunization programs, the most common venue for HPV vaccination in LMICs, therefore also reducing costs by allowing for annual rather than semi-annual school-based campaigns [44]. In response to the growing evidence base for alternative schedules, some policy changes are already being made, such as the recent WHO Strategic Advisory Group of Experts (SAGE) on Immunization advisory that states girls aged 9–10 can receive a two-dose schedule 3–5 years apart [45]. Further evidence, however, is required to inform more widespread changes to current HPV vaccination schedule policies.

### Limitations

The present meta-analysis is subject to a variety of limitations, the most salient of which is a lack of extended interval and single dose data for robust comparisons among HPV-seronegative girls and young women aged 9–26, stratified by vaccine type and time points for assessment of immune response. The lack of extended interval and single dose data in the published literature has several implications for the results of the meta-analysis that may limit the interpretability of the findings. First, relevant data were not available for each vaccine at each time point. For example, no extended interval data were available for two of the licensed vaccines at 72 months post-first dose; thus, the present meta-analysis does not address the long-term comparability of immune response with an extended interval schedule for those vaccines. Further, no extended interval data were available for the 15–26 age range, restricting the present analysis to ages 9–14 years. Single dose data had similar restrictions, with no short-term data and no data for the nonavalent vaccine. Finally, the non-inferiority analysis was based on measures of immunogenicity but not on efficacy. Without a standard titer threshold that correlates with protection, it is difficult to conclude if some of the inferior immunogenicity responses correlate to a lack of vaccine efficacy.

## 5. Conclusions

When evaluated against standard schedules, most extended interval analyses suggested that giving doses farther apart will confer non-inferior immune responses. Single doses did not meet criteria for non-inferiority in any comparisons. For both schedules, sparse data limited the number of possible comparisons. Given the possible large economic and health benefit of extended interval and single dose HPV vaccinations, further research is warranted.

## Figures and Tables

**Figure 1 vaccines-08-00618-f001:**
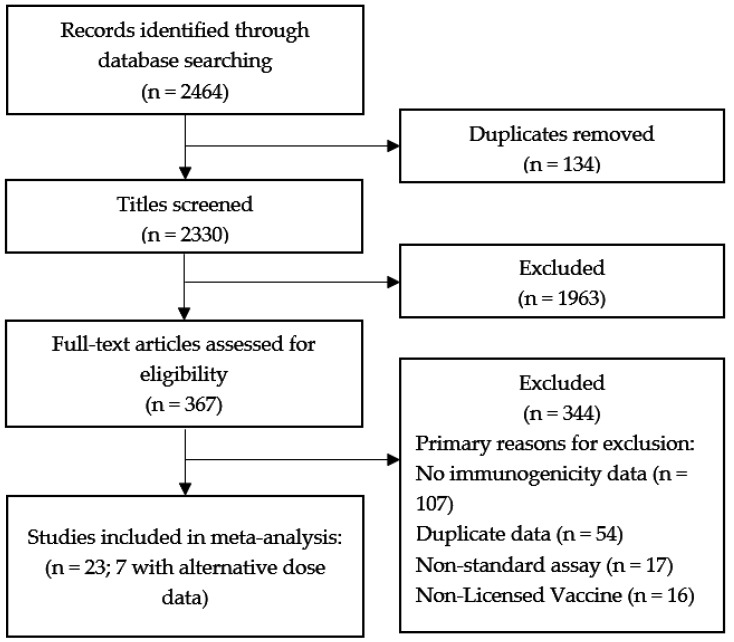
PRISMA diagram.

**Figure 2 vaccines-08-00618-f002:**
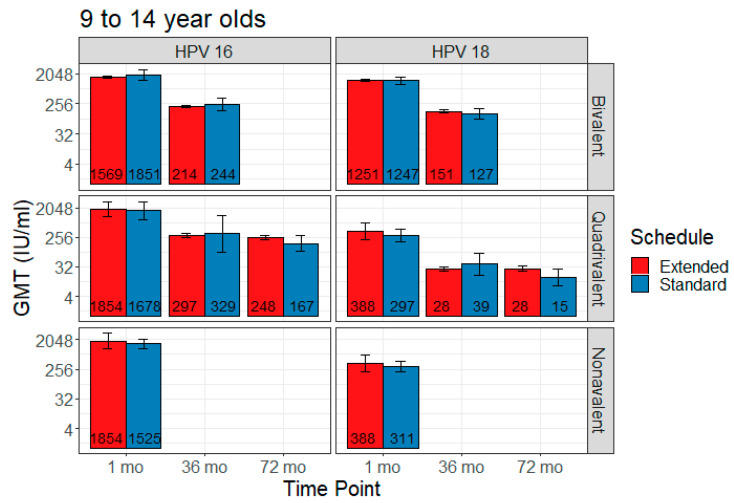
Comparison of pooled geometric mean titer (GMT) for extended interval versus standard schedule (with 95% CI error bars).

**Figure 3 vaccines-08-00618-f003:**
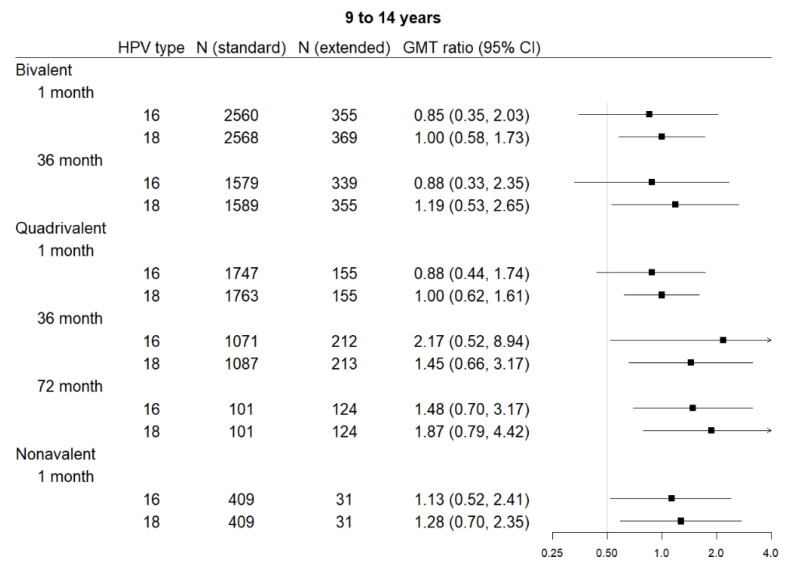
Forest plot of GMT ratio of extended interval versus standard schedule (with 0.5 indicating the non-inferiority cut-point—i.e., those with lower bounds greater than 0.5 are considered to meet non-inferiority criteria), by vaccine, human papilloma virus (HPV) subtype, time point, and age group.

**Figure 4 vaccines-08-00618-f004:**
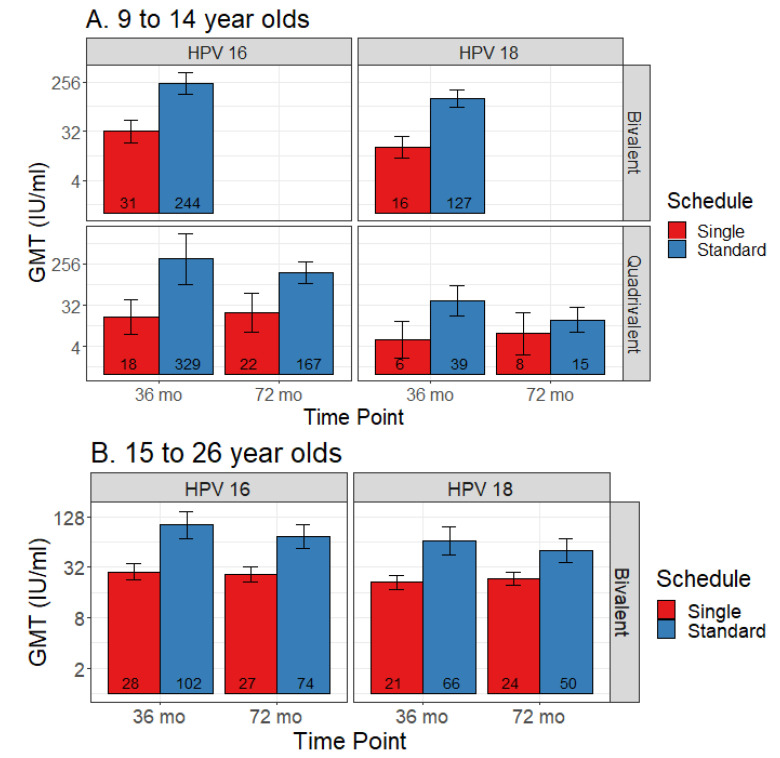
Comparison of geometric mean titers (GMT) for single dose versus standard schedule (with 95% CI error bars) among (**A**) 9 to 14 year olds and (**B**) 15 to 26 year olds.

**Figure 5 vaccines-08-00618-f005:**
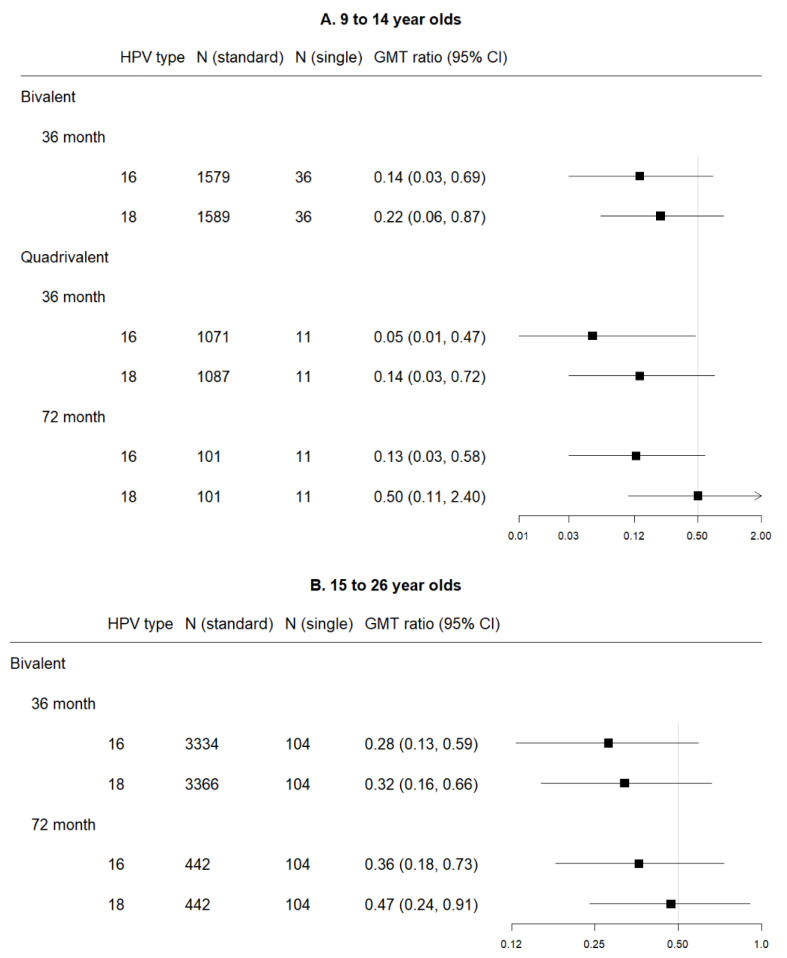
Forest plot of GMT ratio of single dose versus standard schedule (with 0.5 indicating the non-inferiority cut-point—i.e., those with lower bounds greater than 0.5 are considered to meet non-inferiority criteria), by vaccine, HPV subtype, and time point among (**A**) 9 to 14 year olds and (**B**) 15 to 26 year olds.

**Table 1 vaccines-08-00618-t001:** Included single dose and extended interval studies.

Author, Year	Alternative Schedules		Dose Schedules (Months)	Study Design *	Age Range(s)		N **	Vaccine(s)	Region(s)
	Extended	Single			9–14	15–26			
Gilca 2019 [20]	x	x	Single; (0,36–48); (0,60–72); (0,84–96)	NRS	x		31	Quadrivalent, nonavalent	N. America
Huang 2017 [26]	x		(0,1,6); (0,12)	RCT	x	x	669	Bivalent	Asia, Europe, N. America
LaMontagne 2013 [27]	x		(0,2,6); (0,12,24)	RCT	x		223	Quadrivalent	Asia
Lamontagne 2014 [30]		x	Single; (0,2,6)	Cross-sectional	x		231	Bivalent	Africa
Neuzil 2011 [28]	x		(0,2,6); (0,12,24)	RCT	x		418	Quadrivalent	Asia
Puthanakit 2016 [29]	x		(0,1,6); (0,6); (0,12)	RCT	x	x	1195	Bivalent	Asia, Europe, N. America
Safaeian 2018 [10]		x	Single; (0,1,6); (0,6)	RCT		x	330	Bivalent	N. America

* RCT = Randomized control trial; NRS = Non-randomized control trial. ** Sample size for data included in analysis.

## Data Availability

The data that support the findings of this study are available from the corresponding author upon reasonable request.

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
