# Peer review of "Immunogenicity of Alternative Dosing Schedules for HPV Vaccines among Adolescent Girls and Young Women: A Systematic Review and Meta-Analysis"

_vaccines, 2020, doi:10.3390/vaccines8040618_

Round 1

Reviewer 1 Report

A systematic review and meta-analysis was performed to compare data obtained from vaccination schedules with bivalent, quadrivalent and nonavalent HPV vaccines. Non-inferiority was tested for immunogenicity obtained with extended vaccination intervals and single dose schedules versus standard dose schedules.

Studies in girls 9 to 26 years old including geometric mean titers (GMT) for HPV types 16 and 18 following vaccination with the HPV vaccines (bivalent, quadrivalent or nonavalent), were included. Vaccination schedules were standard, extended (at least 12 months between the first and second dose), or single dose, via intramuscular administration. Differential immunogenicity was evaluated by age group (9-14 and 15-26). Non-inferiority was defined as the lower bound of the 95% CI for the GMT ratio being greater than 0.5 and was evaluated through the estimated ratio of pooled GMT comparing extended intervals or single dose with standard schedules.

367 studies were included for full review criteria, but 344 were excluded were excluded because they did not contain the necessary information. Therefore, only 23 studies were analyzed (conducted in North America, South America, Europe, Asia and Africa), where 4 contributed with data of extended interval, 2 to single dose, 1 to both and 16 to comparator data. The sample size of those studies ranged from 18 to 2635.

Studies with bivalent vaccine had GMT data of extended interval for 1 and 36 months post first dose; those with quadrialent vaccine, for 1, 36 and 72 months; and those for nonavalent vaccine, for 1-month post last dose. For the bivalent vaccine GMT ratios showed non-inferiority between extended intervals and standard schedules for HPV18, although non-inferiority was not met for HPV16 at the same intervals. For the quadrivalent vaccine, non-inferiority was shown for HPV18 at 1, 36 and 72 months, while for HPV16, at 36 and 72 months. For the nonavalent vaccine non-inferiority was demonstrated for HPV16 and 18 at 1-month post last dose. For single dose non-inferiority was not met for bivalent or quadrivalent vaccine for both HPV16 and 18 types at all time points.

This is a relevant and well-written work with an adequate design and analysis, which gives information about the feasibility of extending vaccination schedules with a non-inferiority result compared with standard schedules. Although limited in the number of studies that presented the relevant information, non-inferiority was not met for single dose, but the analysis showed that extended schedules could be a good alternative al least for the range or 9 to 14 years old.

As the authors mention, a limitation was that data was not available for all the analyzed time points for the three vaccines and the analysis was based on GMT radios which does not guarantee efficacy.

In general, the discussion is also well conducted.

Therefore, I consider that this manuscript is relevant and deserves to be published.

Author Response

Response to reviewers

Thank you very much for the thorough and thoughtful feedback on our manuscript. Below are changes based on the cumulative feedback for minor English revisions:

Line 40: “1-2” changed to “1 or 2”

Line 42: “0,6” changed to “0, 6 months

Line 43: “quadrivalent vaccine and nonavalent” changed to “quadrivalent and nonavalent vaccines”

Line 68: “of girls aged 9-26 years” changed to “of adolescent girls and young women aged 9-26 years”

Line 81: “estimate” changed to “estimates

Line 101: “recommendation age groups” was dropped

Line 145: “36 month” and “72 month” changed to “36-month” and “72-month”

Line 259: “18” changed to “-18”

Line 276: “all three collected” changed to “all three included studies collected”

Line 291: “extended interval (>12 months) dosing” changed to “extended interval dosing”

Line 292: “currently recommended two-dose schedule” changed to “currently recommended schedules”

Line 320: “immunogenicity, but” changed to “immunogenicity but”

Line 325: “non-inferiority of extended interval dosing was mixed across vaccines, subtypes, and time points” was dropped

Reviewer 2 Report

In this manuscript, the authors have addressed an important public health issue: to review and compare existing data on immunogenicity of extended interval and single dose schedules versus standard dose of currently available FDA and EMA approved HPV vaccines. Evidence of non-inferiority of alternative dosing schedules from previous studies can in fact provide important information in terms of improved HPV vaccination stategies as well as reducing health costs.

The authors have appropriately conducted both a systematic literature search and a meta-analysis of the selected studies, the results of which have been illustrated in a clear and comprehensive manner.

In the discussion, the authors indicate the conclusions that can be made from the meta-analysis as well as appropriately discussing the limitations of their analysis together with the need for further research being warrented in this important public health issue.

In conclusion in my opinion this manuscript addresses an important public health issue as well as being scientifically sound and therefore warrants to be published in the present form.

Author Response

(The authors gave the same response as above.)

Reviewer 3 Report

The manuscript "Immunogenicity of alternative dosing schedules for HPV vaccines among adolescent girls and young women: a systemic review and meta-analysis" is well written and discussed. The manuscript would be of great interest to the reader of the "Vaccines" journal.

Author Response

(The authors gave the same response as above.)
